# A Comparative Study of Approaches to Improve the Sensitivity of Lateral Flow Immunoassay of the Antibiotic Lincomycin

**DOI:** 10.3390/bios10120198

**Published:** 2020-12-03

**Authors:** Kseniya V. Serebrennikova, Olga D. Hendrickson, Elena A. Zvereva, Demid S. Popravko, Anatoly V. Zherdev, Chuanlai Xu, Boris B. Dzantiev

**Affiliations:** 1Research Center of Biotechnology of the Russian Academy of Sciences, A.N. Bach Institute of Biochemistry, Leninsky Prospect 33, 119071 Moscow, Russia; ksenijasereb@mail.ru (K.V.S.); odhendrick@gmail.com (O.D.H.); zverevaea@yandex.ru (E.A.Z.); dspopravko@mitht.ru (D.S.P.); zherdev@inbi.ras.ru (A.V.Z.); 2School of Food Science and Technology, Jiangnan University, Wuxi 214122, China; xcl@jiangnan.edu.cn

**Keywords:** lateral flow immunoassay, antibiotics, lincomycin, gold nanoparticles, quantum dots, surface-enhanced Raman spectroscopy

## Abstract

This study provides a comparative assessment of the various nanodispersed markers and related detection techniques used in the immunochromatographic detection of an antibiotic lincomycin (LIN). Improving the sensitivity of the competitive lateral flow immunoassay is important, given the increasing demands for the monitoring of chemical contaminants in food. Gold nanoparticles (AuNPs) and CdSe/ZnS quantum dots (QDs) were used for the development and comparison of three approaches for the lateral flow immunoassay (LFIA) of LIN, namely, colorimetric, fluorescence, and surface-enhanced Raman spectroscopy (SERS)-based LFIAs. It was demonstrated that, for colorimetric and fluorescence analysis, the detection limits were comparable at 0.4 and 0.2 ng/mL, respectively. A SERS-based method allowed achieving the gain of five orders of magnitude in the assay sensitivity (1.4 fg/mL) compared to conventional LFIAs. Therefore, an integration of a SERS reporter into the LFIA is a promising tool for extremely sensitive quantitative detection of target analytes. However, implementation of this time-consuming technique requires expensive equipment and skilled personnel. In contrast, conventional AuNP- and QD-based LFIAs can provide simple, rapid, and inexpensive point-of-care testing for practical use.

## 1. Introduction

The lateral flow immunoassay (LFIA) is a common analytical platform for the point-of-care testing of medical diagnostics and environmental monitoring because of its rapidity and simplicity. The LFIA provides clear advantages, including the availability of results within a few minutes, the small volume of an analyzed sample, and inexpensive and user-friendly point-of-care testing [1]. The LFIA combines immunochemical reactions with a chromatography principle. It relies on interactions between an analyte and pre-immobilized recognition elements initiated by the addition of a liquid sample. The LFIA result is a signal at the test line generated by a nanodispersed reporter used. Despite all the advantages mentioned above, the widespread use of LFIAs has been limited by their insufficient sensitivity. Significant effort has been devoted to improving LFIA sensitivity, including the use of alternative labels and detectors, as well as the addition of amplification stages [2,3].

To improve the sensitivity of the immunoassay, integration of the LFIA and surface-enhanced Raman spectroscopy (SERS) was proposed. Because of a simple and cost-effective synthesis, gold and silver nanoparticles are the most common SERS substrates [4]. Typically, nanostructured substrates are functionalized with Raman reporter molecules to produce strong and characteristic peaks in SERS spectra, thus enabling quantitative detection of target analytes. The effectiveness of the SERS-based LFIA technique has been confirmed in numerous recent studies [5,6,7,8]. In addition to common AuNPs or latex beads, magnetic and fluorescent particles are used as labels in LFIAs. QDs are used as labels because of their unique optical properties, such as high fluorescence, broad and continuous distributed excitation, photostability, and proven immunoassay effectiveness [9,10]. LFIAs with magnetic and photoluminescent labels showed improved sensitivity for a wide range of analytes [11,12,13,14]. Among other markers applied in LFIA, carbon nanoparticles can be mentioned [15,16]. Compared to other labels, carbon nanoparticles are easily detected visually, which contributes to reducing the detection limit of the analyte.

A survey of the literature shows there have been many works published on new immunoassay markers, but they do not go beyond the description of the effectiveness at detecting a particular analyte or report a comparison of the results with conventional gold nanoparticle-based LFIAs. These regularities are poorly transformed into other objects of research. Therefore, the assessment of the test systems with the same reagents that vary according to the kind of marker and readout technique applied will provide more information.

During the study, we explored three approaches to improving LFIA sensitivity. To verify the effectiveness of the proposed methods, we selected the antimicrobial lincomycin (LIN), which is a product of *Streptomyces lincolnensis* bacteria. The known varieties of methods for quantitative detection of LIN include mainly microbiological and chromatographic techniques [17]. The use of accurate chromatographic methods is a common practice to identify and quantify antibiotics in different matrices. Although chromatography–mass spectrometry is a highly sensitive and efficient method, its use requires sample pretreatment, costly equipment, and specially trained personnel [18,19]. Recently, other techniques have also been reported for the determination of LIN in foodstuffs [20,21]. Numerous studies have reported the use of the enzyme-linked immunosorbent assay (ELISA) and LFIA for monitoring LIN residues [22,23,24]. However, despite the availability of the techniques to control antibiotics, there is great demand for the development of highly sensitive alternative ways of (a) achieving simple pretreatment procedures (reduce it to dilution eliminating the matrix effect) and (b) minimizing the risk of long-term consumption of contaminants at concentrations below threshold levels.

In this study, the same bioreagents were used to compare different labels and readout systems in a competitive LFIA for LIN. An increase in competitive LFIA sensitivity is possible by reducing the concentration of immunoreagents; however, this decrease is limited by the ability to detect the analytical signal. Beyond the optimization of reagent concentrations, improving the signal-generating elements and readout techniques are other effective strategies to achieve increased assay sensitivity. Moreover, the integration of sensitive detection techniques with LFIA allows for a reduction in immunoreagent consumption.

The current study is a systematic investigation using LFIA integrated with different labels (AuNPs and QDs) and readout techniques (colorimetry, fluorescence, and SERS) to detect LIN. AuNPs were implemented both for traditional colorimetric detection and for coupling to SERS readouts. The quantitative detection of LIN was performed by registering the colorimetric or fluorescence intensity of AuNPs or QDs, respectively, captured on the test line. To design a SERS-based LFIA, AuNPs functionalized with 4-mercaptobenzoic acid (4-MBA) and coupled with anti-LIN monoclonal antibodies (AuNPs–MBA–Ab) were used as a SERS reporter bioprobe. In this case, a conventional LFIA procedure was followed by registration of Raman spectra from the test line.

## 2. Materials and Methods

### 2.1. Reactants

Lincomycin hydrochloride monohydrate (LIN), HAuCl_4_, sodium azide, sodium citrate, Tween-20, Triton X-100, and 4-MBA were obtained from Sigma-Aldrich (St. Louis, MO, USA). N-(3-dimethylaminopropyl)-N′-ethylcarbodiimide hydrochloride (EDC) and sulfo-N-hydroxysuccinimide (NHS) were supplied from Fluka (Buchs, Switzerland). Goat antibodies against mouse immunoglobulins (GAMI) were purchased from Arista Biologicals (Allentown, PA, USA). Bovine serum albumin (BSA) was supplied from Eximio Biotec (Wuxi, China). The CdSe/ZnS QDs with an emission peak at 625 nm were obtained from Invitrogen (Catalog No A10200, Thermo Fisher Scientific, Waltham, MA, USA). All other reagents were of analytical grade.

Ultrapure water (Millipore Corporation, Burlington, MA, USA) with resistivity of 418.2 MΩ was used to prepare the AuNPs and their conjugates as well as LIN stock solutions (100 μg/mL). The LFIAs were carried out in 96-well transparent Costar 9018 polystyrene microplates provided by Corning Costar (Tewksbury, MA, USA). Amicon Ultra-0.5 mL Centrifugal Filter (100 K) was purchased from Millipore (Billerica, MA, USA).

### 2.2. Preparation of Monoclonal Anti-LIN Antibodies

A synthesis of the LIN–BSA conjugate and a preparation of anti-LIN antibodies were carried out in accordance with the procedure described in the study by Cao et al. [25].

### 2.3. Synthesis and Characterization of AuNPs

AuNPs with an average diameter of 30 nm and 40 nm were prepared according to the citrate-reduction method [26]. To obtain 30 nm AuNPs, 1 mL of 1% HAuCl_4_ was added to 97.5 mL of ultrapure water and heated to boiling. After that, 1.5 mL of 1% sodium citrate was added immediately to the boiling solution during vigorous stirring. The mixture was left to boil for 25 min and then cooled. The colloidal AuNPs were stored at 4 °C.

To obtain AuNPs with an average diameter of 40 nm, 1.5 mL of 1% sodium citrate was added to 100 mL of boiling 0.01% HAuCl_4_ aqueous solution under rapid agitation. The solution was then boiled for another 15 min and cooled to room temperature.

The transmission electron microscopic (TEM) images were recorded with a JEM-100C electron microscope (JEOL, Tokyo, Japan) operating at 80 kV. The AuNP preparations were applied to 300-mesh grids (Pelco International, Redding, CA, USA) coated with formvar film. The images obtained were analyzed using Image Tool software (University of Texas Health Science Center, San Antonio, TX, USA). UV–vis absorption spectra were obtained through spectrophotometer UV-2450 (Shimadzu, Kyoto, Japan).

### 2.4. Conjugation of Antibodies to AuNPs

Antibody–AuNPs conjugates were prepared according to the previously described technique [27]. Anti-LIN antibodies were dialyzed against a Tris-HCl buffer (10 mM, pH 8.5), and added to AuNPs at a concentration of 10 μg/mL (OD520 = 1). The mixture was incubated for 45 min while stirring at room temperature. BSA in the final concentration of 0.25% was further added to this preparation, followed by stirring for 15 min. The excess reagents were removed by centrifugation at 9500× *g* for 15 min, followed by resuspension of the antibody–AuNPs pellet in Tris buffer (10 mM, pH 8.5) with 1% BSA, 1% sucrose, and 0.1% sodium azide (TBSA).

### 2.5. Conjugation of Antibodies with QDs

Anti-LIN antibodies were dialyzed against a borate buffer (50 mM, pH 8.7). The molar ratio of QDs to anti-LIN antibodies during synthesis was 1:2. Antibodies (300 μL, 0.2 mg/mL), QDs (25 μL, 8 μM), and freshly prepared EDC and NHS solutions (50 μL, 0.8 mM each) were mixed. After incubation for 90 min in a dark place at room temperature, the resulting mixture was purified by centrifugation at 10,000× *g* for 15 min using Amicon Ultra 100 kDa tubes (Billerica, MA, USA).

The centrifugation was repeated four times, and, finally, 14 μL of QDs with a concentration of 4.26 mg/mL was obtained.

### 2.6. Synthesis of the Raman Reporter Bioprobe

To a solution of 40 nm diameter AuNPs (10 mL), 10 μL of 1 mM 4-MBA in ethanol was added [28]. The mixture was incubated for 3 h, followed by centrifugation at 5000× *g* for 15 min. The resulting pellet was resuspended in water.

To prepare the AuNPs–MBA–Ab bioprobe, Au–MBA conjugate and anti-LIN antibodies were adjusted to pH 8.9 with 0.1 M K_2_CO_3_. Anti-LIN antibodies at a concentration of 10 μg/mL were added to 2 mL of Au–MBA and incubated for 3.5 h at room temperature. Then, 50 μL of 10% BSA was added and incubated overnight at 4 °C. After that, the mixture was centrifuged at 9000× *g*, for 10 min. The pellet was resuspended in an equal volume of water and stored at 4 °C.

### 2.7. Preparation of Test Strips

The schemes of three LFIA formats are shown in Figure 1. Test strips were assembled using MdiEasypack membrane sets (Advanced Microdevices, Ambala Cantt, India) comprising the following elements: a plastic support, a CNPC nitrocellulose working membrane with a pore size of 15 μm, a PT–R7 conjugate fiberglass pad (in case of conventional AuNP- and QD-based LFIAs), a GFB-R4 sample pad, and an AP045 absorbent pad. The control line was formed by applying 0.5 mg/mL GAMI in a K-phosphate buffer (PBS, 50 mM, pH 7.4, with 0.1 M NaCl) by an Iso-Flow automatic dispenser (Imagene Technology, Hanover, NH, USA). To form a test line, LIN–BSA conjugate (0.5 mg/mL—for AuNPs-based, 0.15 mg/mL—for QD-based, and 0.2 mg/mL—for SERS-based LFIAs, in PBS) was applied. After that, the test strips were dried at 37 °C for 2 h. For AuNP-based LFIA, the antibody-AuNPs conjugate in TBSA containing 0.05% Tween-20 was applied to the conjugate pad and dried at room temperature overnight. For QD-based LFIA, 1 µL of antibody–QDs conjugate (0.09 µM) in a borate buffer (BB, 0.05 M with 1% BSA, 0.1% sucrose, and 0.1% sodium azide, 0.05% Tween-20) was applied to the interface of the sample pad and nitrocellulose membrane and dried at room temperature overnight. Finally, the assembled multimembrane composites were cut into individual test strips 3 mm wide using an automatic guillotine cutter (Index Cutter-1, A-Point Technologies, Gibbstown, NJ, USA).

### 2.8. LFIA Procedures

#### 2.8.1. Colorimetric and Fluorescent LFIAs

Solutions of LIN (1 µg/mL–1 pg/mL) in PBST (100 μL) were dripped onto the microplate wells. The test strips were vertically placed into the well and left to react for 15 min. The color intensity (in the case of AuNP-based LFIA) of the formed bands was scanned by the CanoScanLiDE 90 (Canon, Tokyo, Japan). The fluorescence intensity (for QD-based LFIA) was recorded under UV light excitation. The obtained images were then digitized using the TotalLab program (Nonlinear Dynamics, Newcastle upon Tyne, UK).

#### 2.8.2. SERS-Based LFIA

To perform the SERS-based LFIA, 2 μL of AuNPs–MBA–Ab bioprobe was pipetted onto the sample pad approximately 1 cm below the nitrocellulose membrane, and 100 μL of LIN (100,000–1 × 10^−8^ ng/mL) in PBS containing 0.05% Triton X-100 (PBST) was added into the microplate wells. The test strips were vertically inserted into the wells and left to react for 15 min. Then, the Raman spectra from 10 points along the middle of the test line were collected using a DXR Raman microscope (Thermo Fisher Scientific, Madison, WI, USA). The SERS settings were selected with the identical registering technique [29,30]. All spectra were obtained under the same conditions: The excitation source was tuned at 780 nm and laser power of 20 mW; the exposure time was 10 s. A 10× objective lens (NA = 0.25) was used to focus a laser spot on the surface of the test strip.

## 3. Results and Discussion

### 3.1. Synthesis and Characterization of Signal Markers

AuNPs were used as a reporter label in conventional and SERS-based LFIAs. AuNPs of a diameter close to 30 nm were reported to be optimal for traditional immunochromatography [31], whereas larger particles are preferable in SERS-based LFIAs. The optimal size of AuNPs for the preparation of a SERS-active probe was previously found to be no more than 50 nm [32]. Therefore, to achieve a desirable sensitivity and high reproducibility in SERS-based LFIAs, AuNPs with an average size of 40 nm were preferred. To prepare AuNPs with diameters of 30 and 40 nm, a simple method of sodium citrate-associated reduction of chloroauric acid was applied. According to this method, the size of the resulting AuNPs is varied by adding different amounts of the reducing agent: to obtain larger particles, a smaller volume of reducing agent is required. The selection of 4-MBA as a Raman reporter molecule stems from the widespread use of thiol-containing aromatic molecules because of their ability to conjugate directly to the gold surface and provide surface carboxyl groups for biomolecule binding [33]. To optimize the composition of the AuNPs–MBA–Ab bioprobe, the amount of added 4-MBA was varied in the range of 10–50 µL per 10 mL of AuNPs. It was demonstrated that an excess of 4-MBA could cause the aggregation of AuNPs, as evidenced by a red-shifted absorbance peak in UV–vis spectra and a color change of the AuNPs–MBA–Ab probe (data not shown). Therefore, 10 µL of 1 mM MBA was proven to be sufficient for preparation of a stable AuNPs–MBA conjugate.

The size and shape of AuNPs were estimated by TEM and UV–vis spectroscopy (Figure 2). The as-prepared AuNPs showed localized surface plasmon resonance at 523 and 527 nm for AuNPs of 30 and 40 nm, respectively. After the conjugation process, the slight redshift of the maximum peak of AuNPs–MBA–Ab was observed, which indicates a successful conjugation of Au–MBA and anti-LIN antibodies. The TEM images revealed spherical morphology and homogeneity with a size distribution in the range of 29.5 ± 7.4 nm and 39.5 ± 5.0 nm, and a degree of ellipticity of 1.3 for two AuNPs preparations (Figure 2b,c). According to the manufacturer, the size of carboxyl quantum dots varies from 15 to 20 nm.

### 3.2. AuNP-Based LFIA

Given the need to detect a low molecular weight compound in this study, a direct competitive LFIA format was performed (Figure 1). In this assay, free LIN present in the sample competed with the immobilized LIN–BSA conjugate in regard to binding with specific anti-LIN antibodies. The binding sites of the specific anti-LIN antibodies labeled with different nanodispersed markers were first occupied with the target analyte. And thereafter the excess labeled antibodies were captured by the LIN-BSA conjugate, which in turn was detected by employing different detection techniques. Thus, the signal intensity on the test line of the strip was inversely proportional to the concentration of LIN in the sample. The preliminary characterization of the immune properties of monoclonal anti-LIN antibodies used by ELISA confirmed their high affinity (Appendix A) and allowed for the development of LFIAs.

The scheme of the conventional AuNP-based LFIA is presented in Figure 1a. For the LFIA, LIN–BSA conjugate, and GAMI were applied to form test and control lines on the working membrane, respectively. The specific antibody-labeled AuNPs was immobilized on the fiberglass pad. The assay conditions were optimized to achieve the lowest detection limit at a high amplitude of the analytical signal. As a result, the following conditions were found to be optimal for three formats of assay: 0.5 mg/mL—for AuNP-based, 0.15 mg/mL—for QD-based, and 0.2 mg/mL—for SERS-based LFIAs (the concentration varies from 0.2 to 1 mg/mL) and 0.5 mg/mL for GAMI (the concentration varies from 0.15 to 0.5 mg/mL). The AuNPs-anti-LIN antibodies solution was then applied to the conjugate pad at the concentration corresponding to OD_520_ = 1 (we tested OD_520_ in the range from 0.5 to 2.5). The overall performance of the LFIA was explored by varying the concentration of the analyte (from 1000 to 0.001 ng/mL). Under optimal experimental conditions, the AuNP-based LFIA exhibits linearity over the range of 0.7–7.2 ng/mL with an instrumental detection limit of 0.4 ng/mL (Figure 3). The cutoff was 10 ng/mL with the assay duration of 15 min.

### 3.3. QD-Based LFIA

The scheme of QD-based LFIA is demonstrated in Figure 1b. For QD-based LFIA, the selection of working membranes aimed to decrease background fluorescence was carried out together with the optimization of specific reagent concentrations described above. For this purpose, CNPC SS12 12/15 µ (Advanced Microdevices), HF120, and HF180 (Millipore) membranes differing in pore size and flow rates were tested. Figure 4 indicates that the use of CNPC SS12 (of a 12 and 15 µm pore size, respectively) leads to the formation of background coloration over its entire surface. Testing of Millipore membranes with different pore sizes and flow rates demonstrated that the application of the Millipore HF180 membrane facilitated achieving the maximum analytical signal intensity (as opposed to a 20% reduction in intensity when using a HF120 membrane), eliminating nonspecific binding, and ensuring uniform movement of samples.

The next stage of assay optimization was to select the optimal reaction medium that would decrease nonspecific binding and provide a higher signal intensity. The use of PBST as a buffer solution for QD-based LFIA led to nonspecific binding of the antibody–QDs conjugate and background staining of the working membrane. For LIN detection, significantly higher signal intensities were obtained with BB. It is acknowledged that, for the effective elution of the antibody–QDs conjugate and its movement along the membrane, detergents must be added to the buffer [26]. It was shown that the addition of Tween-20 (0.05%), BSA (1%), and sucrose (0.1%) to BB eliminated the nonspecific sorption of QD-labeled antibodies in the test zone and increased the intensity of the analytical signal by 15%. BSA and sucrose were added to the buffer to reduce the flow rate (due to a viscosity increase) and, hence, to maximize the contact time of the sample with the labeled antibodies. Furthermore, the use of BSA allows blocking the sites of nonspecific sorption of the conjugate [34].

An antibody–QDs conjugate solution was applied to the interface of the sample pad and working membrane at a volume of 1 µL and a concentration range of 0.09 to 0.28 µM. The 0.09 µM conjugate concentration was shown to give the optimal fluorescence intensity. A further decrease in the concentration of the antibody–QDs conjugate led to a drop in the signal amplitude and a decrease in the reproducibility of the test results.

Figure 5 shows the calibration curve for LIN detection in the optimized LFIA. The instrumental LOD was 0.2 ng/mL and the dynamic linear range was 0.6–10.4 ng/mL. The visual LOD was 20 ng/mL. QD-based LFIA can provide results in 15 min.

### 3.4. SERS-Based LFIA

The principle of the SERS-based LFIA is illustrated in Figure 1c. In this study, the bioconjugates of the anti-LIN antibody and AuNPs functionalized with 4-MBA were applied as both a Raman reporter bioprobe and a detection probe. In this study, 4-MBA was chosen as the reporter molecule because of its ability to provide a strong binding to the AuNP surface and high SERS-signal while being in proximity with a metal surface (enhancement factor up to 1 × 10^7^) [35]. To obtain the optimal characteristics of the test-system, the following parameters were optimized: The amount of the LIN–BSA conjugate immobilized on the test line; the amount of the Raman reporter bioprobe. After immersing the test strips in 100 μL of the LIN solutions, visual staining was detected on the test line after 15 min. According to the competitive format of assay, the color intensity and, consequently, the SERS signal provided by the AuNPs-MBA-Ab probe is inversely proportional to the LIN concentration. As shown in Figure 6, the SERS spectra of the Raman reporter bioprobe are characterized by two intense peaks at 1077 cm^−1^ and 1580 cm^−1^, which in conformity with spectral data for the MBA molecule [36] correspond to vibrations in the aromatic ring. Therefore, this confirms the specific binding of reporter bioprobe to the LIN-BSA conjugate on the test line. The highest peak at 1077 cm^−1^ was used further to quantify the antibiotic content. As follows from the spectra corresponding to different LIN concentrations, the SERS intensity gradually decreases at 1077 cm^−1^ with an increase in the concentration of antibiotics.

To investigate the impact of the Raman reporter bioprobe, AuNPs–MBA–Ab in amounts ranging from 1 to 5 µL were dotted on a sample pad of the strip (Figure 7a). For the coating antigen immobilized on the test line, LIN–BSA with concentrations ranging from 0.2 to 0.5 mg/mL were investigated, respectively (Figure 7b). The amount of these components is shown to have no significant influence on assay sensitivity (Figure 7).

When the LIN–BSA concentration and a loading of an AuNPs–MBA–Ab bioprobe increased, the Raman intensities decreased. In contrast, a decrease in the amount of Raman reporter bioprobe and the concentration of the immobilized LIN–BSA conjugate allowed for the expansion of the dynamic range of the detected LIN concentrations. Therefore, the optimal amount of LIN–BSA conjugate was 0.2 mg/mL and the amount of AuNPs–MBA–Ab bioprobe added to the strip was 2 μL. The results shown in Figure 8 indicate that the dynamic linear range of the SERS-based LFIA varies from 2.8 × 10^−6^ to 10 ng/mL with a detection limit of 1.4 × 10^−6^ ng/mL. Such improved analytical characteristics can be explained by the high sensitivity of SERS detection toward a reporter molecule, which allows for simultaneous reduction in the amount of immobilized LIN-BSA conjugate and the Au-MBA-Ab reporter bioprobe. Not enough attention is being paid to the development of LFIA for LIN. According to the data shown in Table 1, most studies are devoted to the development of conventional colorimetric LFIAs for the detection of LIN, including instances where LIN is part of a panel of antibiotics (multiplex assay format). 

Replacing AuNPs with fluorescent microspheres made it possible to achieve a small gain in the assay sensitivity [23]. However, a significant decrease in the detection limit of LIN was achieved in our previous study (up to 8 pg/mL) when an indirect LFIA was implemented. In this study, the implementation of SERS readout technique in AuNP-based LFIA using the same immunoreagents revealed an approximately three orders of magnitude improvement in assay sensitivity. It should, however, be pointed out that such high sensitivity is resulted not only from the implementation of effective readout techniques but also from the excellent characteristics of the applied antibodies, the affinity of which was 1.15 × 10^9^ M^−1^ (according to information provided by the manufacturer). To date, the integration of immunoassay, in particular LFIA, with the SERS detection technique for the development of highly sensitive quantitative test systems is just getting started. Nevertheless, a number of studies prove the effectiveness of this method for the determination of target analytes at low concentrations [30,40,41,42]. The current study demonstrates excellent performance of the test system reached by the integration of the SERS technique with LFIA, and this integration may be considered as a potential tool for sensitive screenings of antibiotics. However, SERS technique, which is attractive because of its ability to detect extremely low analyte concentrations, is difficult to classify as point-of-care testing unless it is a handheld Raman reader format. As a result of the steps taken to design simple and portable SERS-based LFIA readers, the commercial availability of such devices may be expected in the future [43,44]. On the contrary, AuNP- and QD-LFIAs are fast and cheap out-of-laboratory techniques available today, the results of which can be quantified even using smartphones or handheld readers [45,46]. Therefore, the choice of appropriate technique is influenced by the object of study.

## 4. Conclusions

In the current study, several LFIA approaches using the antibiotic LIN as the relevant contaminant of food products were performed and compared, including conventional AuNP-based LFIA, fluorescent QD-based LFIA, and SERS-based LFIA. AuNP- and QD-based LFIAs are confined to the limit of detection of the nanodispersed label used and the analyses were carried out in the direct competitive format with use of anti-LIN antibodies labeled with AuNPs or QDs. The colorimetric AuNPs-based LFIA was characterized by the detection limit of 0.4 ng/mL. The replacement of the colorimetric marker with a fluorescent one resulted in a slight enhancement in sensitivity (the detection limit was 0.2 ng/mL). To address current challenges of LFIA biosensors associated with the lack of sensitivity and limits in quantitative analysis, the novel SERS-based LFIA for LIN was developed. The limit of detection determined by SERS experiments was 1.4 × 10^−6^ ng/mL. Notably, the sensitivity of AuNP- and QD-based LFIAs are defined by the detection limit of the nanodispersed marker on the test strip, while in the case of SERS-based LFIA an indirect registration of the signal from the Raman reporter molecule using a highly sensitive device is performed. Therefore, the ordinary comparison of the detection limits achieved using the considered three approaches is not quite legitimate and the choice of a nanodispersed marker and a signal detection technique should be determined by several parameters, in particular, the aim of the study, facilities of the laboratory, the nature of the target analyte and requirements to its maximum residue limits. The proposed SERS-based LFIA, which possesses both high sensitivity and quantitative evaluation capabilities, confirms the effectiveness of the SERS technique for the sensitive detection of target analytes. This implies that handheld Raman readers for quantitative LFIA could potentially facilitate sensitive point-of-care tests. To our knowledge, ours is the first report of quantitative LIN detection by fluorescence and SERS-based LFIA, which also presents a promising tool for other contaminants.

## Figures and Tables

**Figure 1 biosensors-10-00198-f001:**
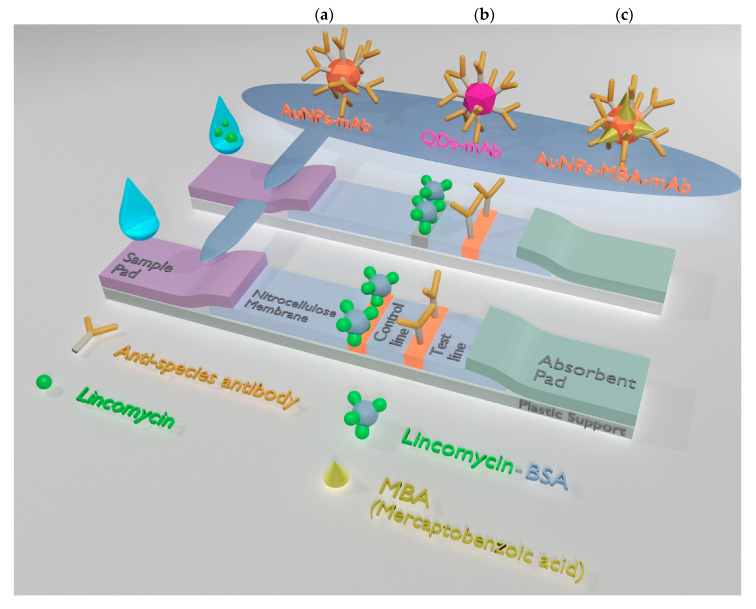
Schemes of lateral flow immunoassay (LFIA) formats developed in the study: conventional colorimetric gold nanoparticle (AuNP)-based LFIA (**a**); fluorescent quantum dot (QD)-based LFIA (**b**); AuNP-based LFIA with surface-enhanced Raman scattering (SERS) detection (**c**).

**Figure 2 biosensors-10-00198-f002:**
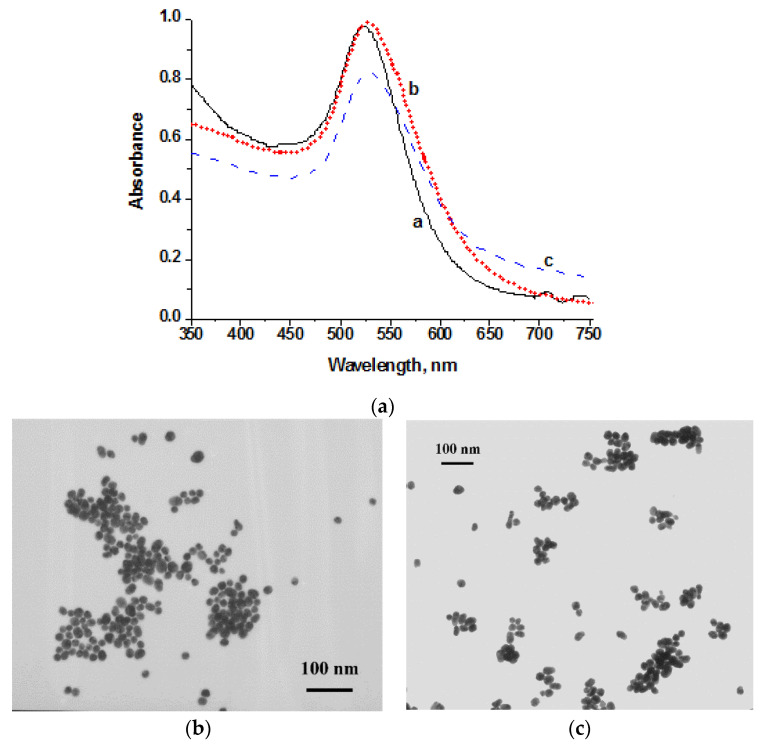
(**a**) UV–vis spectra of AuNPs of 30 nm (**a**) and 40 nm (**b**) diameter, and AuNPs functionalized with 4-mercaptobenzoic acid (4-MBA) and coupled with anti-lincomycin (LIN) monoclonal antibodies (AuNPs–MBA–Ab) (**c**); (**b**) microphotograph of AuNPs for colorimetric LFIA. The average diameter is 29.5 ± 7.4 nm and the degree of polydispersity is 1.3; (**c**) microphotograph of AuNP–MBA for SERS-based LFIA. The average diameter is 39.5 ± 5.0 nm and the degree of polydispersity is 1.3.

**Figure 3 biosensors-10-00198-f003:**
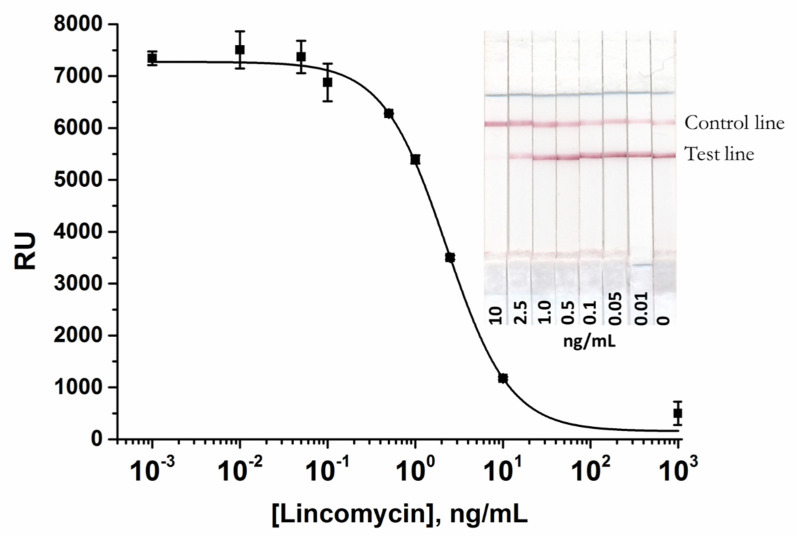
Calibration curve of LIN in the AuNP-based LFIA and the digital photographs of the LFA strips after conventional AuNP-based LFIA procedure. The LIN-BSA conjugate was applied at the test line at a concentration of 0.5 mg/mL. The AuNPs-anti-LIN antibodies solution was applied at a concentration corresponding OD_520_ = 1. LIN concentrations are given at the bottom of the test strips. The error bars indicate the standard deviations for three measurements.

**Figure 4 biosensors-10-00198-f004:**
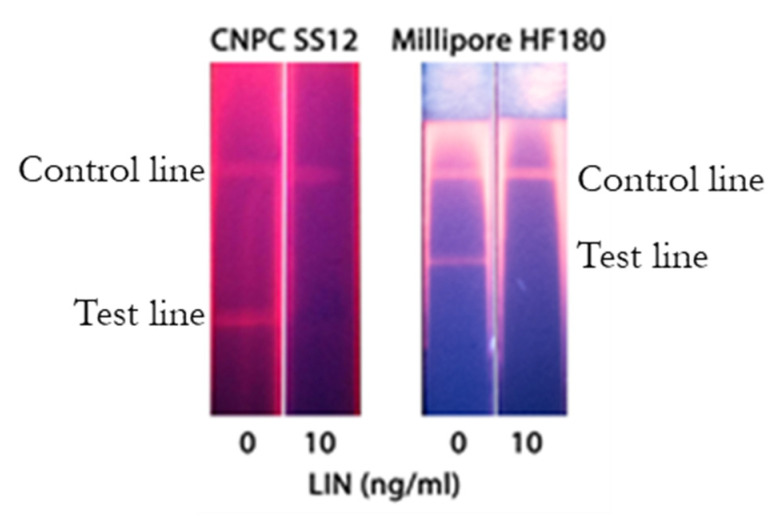
Images of the test zones after the LFIA performed using CNPC SS12 and Millipore HF180 membranes. The LIN-BSA conjugate concentration was 0.15 mg/mL.

**Figure 5 biosensors-10-00198-f005:**
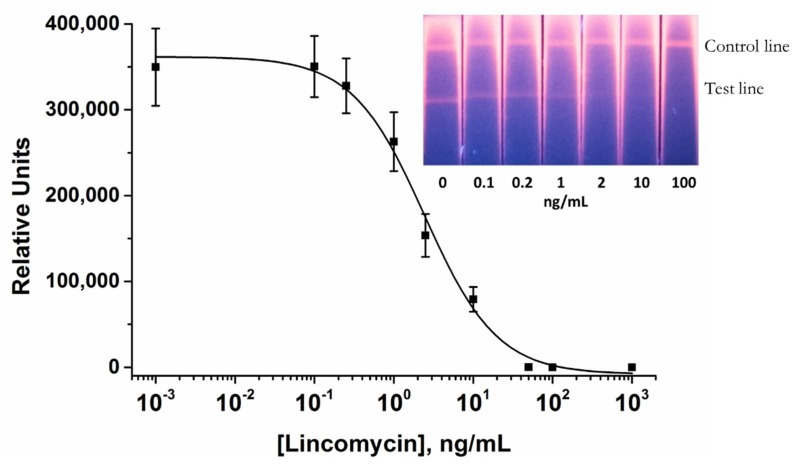
Calibration curve of LIN in the QD-based LFIA and image of the test strips with increasing concentration of LIN ranging from 0.1 to 100 ng/mL under following optimal conditions: The LIN-BSA conjugate was applied at the test line at a concentration of 0.15 mg/mL; 1 µL of antibody–QDs conjugate (0.09 µM) was applied to the interface of the sample pad and nitrocellulose membrane. The error bars indicate the standard deviations for three measurements.

**Figure 6 biosensors-10-00198-f006:**
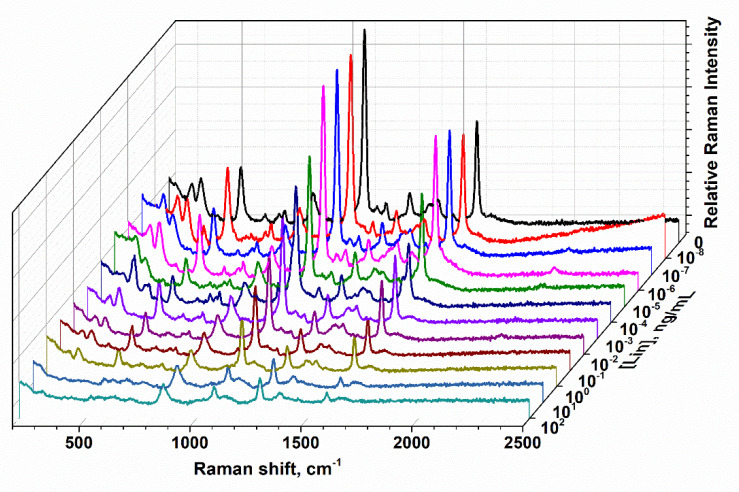
SERS spectra arising from MBA on a test line for various LIN concentrations following the LFIA procedures under optimal conditions: The amount of AuNPs–MBA–Ab bioprobe was 2 μL; the amount of LIN–BSA conjugate was 0.2 mg/mL.

**Figure 7 biosensors-10-00198-f007:**
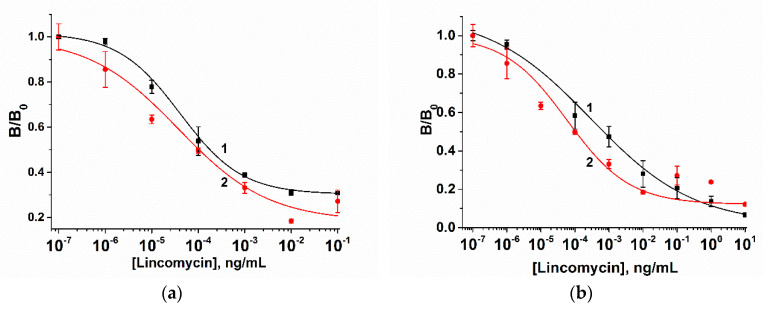
Calibration curves of the SERS-based LFIA of LIN for various LIN–BSA conjugate and AuNPs–MBA–Ab bioprobe amounts. (**a**) The amount of AuNPs–MBA–Ab bioprobe was 1 µL (1) and 4 µL (2); (**b**) the amount of LIN–BSA conjugate was 0.2 mg/mL (1) and 0.5 mg/mL (2). B and B_0_ (B_0_ ≈ 1000 a.u.) correspond to the SERS intensities of MBA at 1077 cm^−1^, when standard and zero LIN solutions were applied to the sample pad, respectively. The error bars indicate the standard deviations for three measurements.

**Figure 8 biosensors-10-00198-f008:**
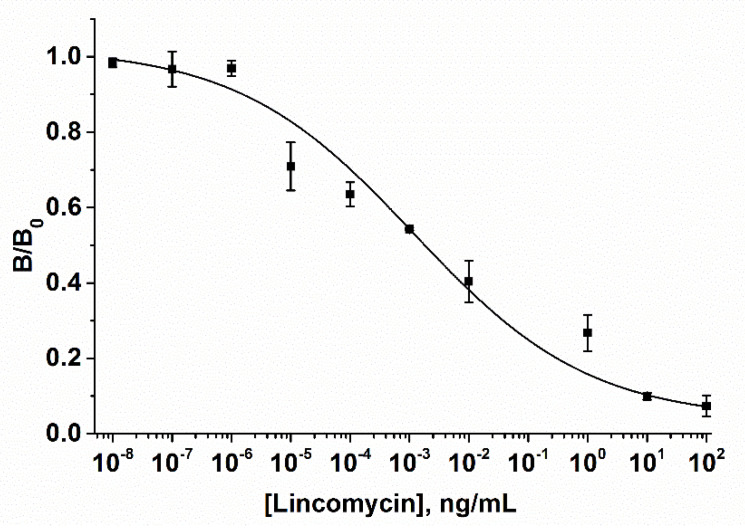
The calibration curve of the SERS-based LFIA for LIN detection according to the following optimal assay conditions: The amount of AuNPs–MBA–Ab bioprobe was 2 µL; the LIN–BSA conjugate concentration was 0.2 mg/mL. The error bars indicate the standard deviation of the Raman intensities from MBA reporter molecule at 1077 cm^−1^ measured from 10 points along the middle of the test line.

**Table 1 biosensors-10-00198-t001:** Lateral flow immunoassay (LFIA) tests for lincomycin detection.

Target Analyte	LFIA Formats	Signal Marker	Limit of Detection	References
Lincomycin (and chloramphenicol, tetracycline)	Multiplex LFIA	30 nm AuNPs	0.4 ng/mL	[37]
Lincomycin (and gentamicin, kanamycin, streptomycin, neomycin)	Multiplex LFIA	15 nm AuNPs	2.5 ng/mL	[38]
Lincomycin	Fluorescence LFIA	Fluorescent microspheres	0.69 ng/mL	[23]
Lincomycin (and clindamycin, pirlimycin)	Conventional LFIA	20 nm AuNPs	10 ng/mL	[39]
Lincomycin	Indirect LFIA	30 nm AuNPs	8 pg/mL	[24]

Abbreviations: AuNPs—gold nanoparticles.

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
