# Peer review of "A Comparative Study of Approaches to Improve the Sensitivity of Lateral Flow Immunoassay of the Antibiotic Lincomycin"

_biosensors, 2020, doi:10.3390/bios10120198_

Round 1
Reviewer 1 Report
The Manuscript by Serebrennikova et al. assesses various detection techniques for lincomycin sensing on the lateral flow immunoassay platform. The study design is straightforward, and the Manuscript is easy to follow. The SERS results look very promising.
Papers with side-by-side comparisons of different approaches are rare, and this fact alone merits the publication.
The Referee would like to share the following comments on the Manuscript:
- SERS is a super-sensitive method of detection. As expected, fluorescence detection outperforms colorimetric assay. The exact detection limits, however, are determined by the SNR of the imaging devices. It is worth noting that the DXR Raman microscope is not just an easy to use point-and-shoot device but is also a very sensitive one. To this end, the comparison of the ThermoFisher microscope with an unlisted imaging device for fluorescence detection does not seem fair (“The fluorescence intensity (for QD-based LFIA) was recorded under UV light excitation.”). For instance, the emission spectrum of QDs is narrow (FWHM ~25 nm). It could be efficiently isolated with proper emission filters, to the effect of a considerable increase in the signal-to-background ratio.
- The Referee urges the authors to amend the text and conclusions accordingly so that the reader will not be misguided by comparing data from the research-grade Raman microscope for SERS modality with the data obtained on less accurate instruments/setups for colorimetric and fluorescence assays.
- The figure captions (Figs 3,5,7,8) lack description of the statistics used, i.e., measures of center, the meaning of whiskers, number of replicas
- Please provide additional technical details in order to improve the reproducibility of the study. Specifically:
– add info on the objective used with DXR for test line measurements
– add all the info on fluorescence image acquisition
– please include catalog numbers for the QDs
- The Referee invites the Authors to share, whenever possible, ‘data not shown’ pieces as supplementary figures/notes.
- It is recommended to provide more technical details within the figure legends (Figs 2-8)
7.The detection limit for fluorescence-based readout in the text (0.2 ng/mL) differs from the one (0.3 ng/mL) referenced in the TOC graphics. Please check
Reviewer 2 Report
This manuscript by Dzantiev and coworkers compares the sensitivity of lateral flow immunoassay for lincomycin detection. The assays include colorimetric (using gold nanoparticles), fluorescence (using quantum dots) and a surface-enhanced Raman spectroscopy (SERS)-based method. The three assays provide lincomycin detection limits of 0.4 ng/mL, 0.3 ng/mL and 1.4 x 10-6 ng/mL, respectively. The authors conclude that the proposed SERS-based LFIA method possesses both high sensitivity and quantitative evaluation capabilities, and that handheld Raman readers for quantitative LFIA could potentially facilitate sensitive point-of-care tests.
The LOD reported for the SERS-based assay is 3-orders of magnitude greater than an indirect LFIA method reported previously by this group (8 pg/mL) in reference 24 (Table 1). All other LOD values for lincomycin detection by other groups are in the ng/mL regime, as observed in this study for the colorimetric and fluorescence assays. What is the binding affinity of lincomycin for the antibody? (This should be included in the manuscript). The SERS detection is based on the changes in the spectra presented in Figure 6 as a function of lincomycin concentration. Prior to addition of lincomycin the SERS spectrum shows 2 major peaks at 1077 cm-1 and another peak at ~ 1500 cm-1. These two peaks should be characterized (what are these peaks due to?). The first 3 additions of lincomycin (red,blue,pink) show virtually no impact on the magnitude of the two peaks. However, upon addition of 10-5 ng/mL lincomycin (green line) the intensity of both peaks clearly decreases. From that point on addition of lincomycin diminishes the intensity of the two peaks. The authors should carefully outline the nature of the Raman signals and why addition of lincomycin causes signal loss.
Other point:
1: Figure 3, describe the insert test strips and the different bands should be properly labeled.
Reviewer 3 Report
The paper, entitled "A Comparative Study of Approaches to Improve the Sensitivity of Lateral Flow Immunoassay of the Antibiotic Lincomycin", is presented with a fair experimental structure and interesting results.
As for me, I have only a few comments before to consider it suitable for publication on this journal.
- Which microscope objective is used for SERS analysis? I find these parameters to be reasonable if you are using a 10x objective with a low NA. The measurement parameters (20 mW and 10 s) are inappropriate for a SERS analysis, even more so if you are using a thiol, which is a stable Raman reporter and which forms a uniform monolayer on gold nanoparticles.
- Is it possible to add a detailed sketch that illustrates and better describes how the test strips work and how they are functionalized, in the three cases already illustrated in Figure 1? It may be of help to those readers who are not familiar with this method.
- The graphs in Figures 3, 5, 6 and 8 should be better described in the caption.
- The caption of figure 6 should specify that the SERS spectra refer to the 4-MBA.
- It would be appropriate to add the 4-MBA SERS intensities to have a numerical estimate of the variation of intensities as a function of Lincomycil concentration (figure 6).
- In addition, has an estimate been made of the enhancement factor (EF) obtained with 4-MBA on AuNPs?
Round 2
Reviewer 2 Report
The authors have adequately addressed the concerns raised by the reviewers in this revised manuscript. This reviewer is now in favor of publication.